Clinical and perinatal outcomes of fresh single-blastocyst-transfer cycles under an early follicular phase prolonged protocol according to day of trigger estradiol levels

Ying Yingfen 1
Lu Xiaosheng 1
Zhang Huina 1
http://orcid.org/0000-0002-6158-3436 Arhin Samuel Kofi 2
Hou Xiaohong 1
Wang Zefan 1
Wu Han 1
Lu Jieqiang 1 jieqianglu@126.com
Tang Yunbing 1 tyb953018@163.com
1 Department of Obstetrics and Gynecology, The Second Affiliated Hospital and Yuying Children’s Hospital of Wenzhou Medical University , Wenzhou, Zhejiang , China
2 School of Allied Health Sciences, University of Cape Coast, PMB , Cape Coast , Ghana
Park Eun-Jung
Electronic publication date: 2021 Jul 26
Publication date: 2021
Volume: 9
Electronic Location ID: e11785
Received 2020 Sep 25; Accepted 2021 Jun 24
Copyright: © 2021 Ying et al.
Copyright year: 2021
Copyright holder: Ying et al.
License: This is an open access article distributed under the terms of the Creative Commons Attribution License, which permits unrestricted use, distribution, reproduction and adaptation in any medium and for any purpose provided that it is properly attributed. For attribution, the original author(s), title, publication source (PeerJ) and either DOI or URL of the article must be cited.
License URL: https://creativecommons.org/licenses/by/4.0/

Keywords: Fresh single blastocyst transfer, Early follicular phase prolonged protocol, Estradiol level, Perinatal outcome, Live birth

Funding: General Projects of the Zhejiang Medical and Health Science and Technology Plan 2021KY211 This research was supported by the General Projects of the Zhejiang Medical and Health Science and Technology plan (No: 2021KY211). The funders had no role in study design, data collection and analysis, decision to publish, or preparation of the manuscript.

==============================
Backgroud

This study’s objectives were to compare the clinical, perinatal, and obstetrical outcomes of patients with different estradiol (E2) levels in fresh single-blastocyst-transfer (SBT) cycles under an early follicular phase prolonged regimen on the day of trigger.

Methods

We recruited patients in fresh SBT cycles (n = 771) undergoing early follicular phase prolonged protocols with β-hCG values above 10 IU/L between June 2016 and December 2018. Patients who met the inclusion and exclusion criteria were divided into four groups according to their serum E2 level percentages on the day of trigger: <25th, 25th–50th, 51st–75th, and >75th percentile groups.

Results

Although the rates of clinical pregnancy (85.57% (166/194)), embryo implantation 86.60% (168/194), ongoing pregnancy (71.13% (138/194)), and live birth (71.13% (138/194)) were lowest in the >75th percentile group, we did not observe any significant differences (all P > 0.05). We used this information to predict the rate of severe ovarian hyperstimulation syndrome (OHSS) area under the curve (AUC) = 72.39%, P = 0.029, cut off value of E2 = 2,893 pg/ml with the 75% sensitivity and 70% specificity. The 51st–75th percentile group had the highest rates of low birth weight infants (11.73% (19/162), P = 0.0408), premature delivery (11.43% (20/175), P = 0.0269), admission to the neonatal intensive care unit (NICU) (10.49% (17/162), P = 0.0029), twin pregnancies (8.57% (15/175), P = 0.0047), and monochorionic diamniotic pregnancies (8.57% (15/175); P = 0.001). We did not observe statistical differences in obstetrics complications, including gestational diabetes mellitus (GDM), gestational hypertension, placenta previa, premature rupture of membranes (PROM), and preterm premature rupture of membranes (PPROM).

Conclusion

We concluded that serum E2 levels on the day of trigger were not good predictors of live birth rate or perinatal and obstetrical outcomes. However, we found that high E2 levels may not be conducive to persistent pregnancies. The E2 level on the day of trigger can still be used to predict the incidence of early onset severe OHSS in the fresh SBT cycle.

Introduction

Assisted reproductive technology (ART), which largely utilizes controlled hyperstimulation (COH), has made the dream of parenthood a reality for many infertile patients (Hilbert & Gunderson, 2019). However, the use of exogenous gonadotropin (Gn) during the COH cycle can often cause serum estradiol (E2) levels to be higher than during the natural cycle (Li et al., 2019). High E2 concentration in the follicular phase causes early onset of the luteinizing hormone (LH) peak, early endometrial luteinization, and a reduced embryo implantation rate (Fatemi & Popovic-Todorovic, 2013). A previous study found that live birth rates were not affected by E2 levels on the day of trigger during the pituitary down-regulation cycle (Huang et al., 2014).

Gonadotropin releasing hormone agonist (GnRH-a) remains the most commonly-used peptide hormone in COH during ART treatments because it can obtain the most oocytes, prevent early-onset LH peaks, reduce luteinization, and improve cycle completion rates (Haydardedeoğlu & Kılıçdağ, 2016). Patients that have been treated with long-acting GnRH-a during COH have shown good compliance and clinical pregnancy rates (Gao et al., 2014; Liao et al., 2015). Our previous study also showed encouraging clinical outcomes for long-acting GnRH-a, regardless of the patient’s menstrual cycle time (Ying et al., 2019).

The ultimate goal of ART is to deliver a single, healthy, living baby as quickly and economically as possible. Single-blastocyst-transfer (SBT) can prevent high incidences of complications such as multiple pregnancies and ovarian hyperstimulation syndrome (OHSS) which results in the continuous emergence of maternal and fetal related complications such as gestational hypertension, premature delivery and low-weight newborns (Zeng & Li, 2019).

SBT clinical outcomes are better during the fresh cycle than clinical outcomes during the fresh cleavage stage transplantation (Glujovsky et al., 2016). Moreover, a recent meta-analysis suggested that in patients undergoing in vitro fertilization/intracytoplasmic sperm injection (IVF/ICSI) cycles, cryopreserved SBT may not be the best choice when compared to fresh SBT (Zeng & Li, 2019).

Until now, the application of fresh SBT under an early follicular phase prolonged regimen has not been thoroughly examined. Do different E2 levels on the day of trigger influence the live birth rate or increase maternal and fetal complications? In this study, we compared the clinical, obstetrical, and perinatal outcomes of using fresh D5 SBT under an early follicular phase long-acting GnRH-a protocol based on our analysis of E2 levels on the day of trigger.

Materials & methods

Participants

This was a retrospective study. We collected data from 1,897 cycles of prolonged protocol between June 2016 and December 2018 at a single reproductive center. We received written informed consent from all participants. Blastocysts from 759 cycles had undergone cryopreservation, three patients canceled their cycles for personal reasons, and 364 cycles showed β-hCG values lower than 10 IU/L on the 12th day following blastocyst transfer. Ultimately, we used data from 771 prolonged regimen cycles that used fresh SBT and had β-hCG values of at least 10 IU/L (Fig. 1).

Figure 1 Flow chart.

A total of 1,897 cycles of prolonged protocol were collected during the period from Jun. 2016 to Dec. 2018 in our single reproductive center. Among them, 759 cycles underwent cryopreservation of all blastocysts, three cycles were cancelled for personal reasons, and 364 cycles’ β-hCG value was less than 10 IU/L on the 12th day after blastocyst transfer. Finally, 771 cycles of fresh SBT of prolonged regimen were obtained and their β-hCG value was at least 10 IU/L. PERCENTILE function of Microsoft Excel was used to get the corresponding percentile of E2 value. They were <25th percentile Group (E2 level: 212–1,677 pg/ml, n = 194), 25th–50th percentile Group (E2 level: 1,680–2,380 pg/ml, n = 195), 51st–75th percentile Group (E2 level: 2,407–3,028 pg/ml, n = 188) and >75th percentile Group (E2 level: 3,036–6,526 pg/ml, n = 194). Statistical analysis was used to compare patients’ data.

The inclusion criteria were as follows: (1) all candidates were between the ages of 20 and 42 years old and had infertility due to tubal obstruction, endometriosis (EMs), polycystic ovary syndrome (PCOS), or ovarian polycystic changes confirmed by transvaginal ultrasound, male factors, or idiopathic reasons; (2) the endometrial double layer thickness on the day of trigger was greater than 6 mm; (3) the serum progesterone (P4) level did not exceed 1.98 ng/ml on the day of trigger; and (4) they had one or more D5 3BC blastocyst.

The exclusion criteria were as follows: (1) patients required all blastocysts to be cryopreserved on the day of fresh SBT treatment; (2) endometrial adhesion, submucosal myoma, or uterine diameters longer than 65 mm; (3) obvious infection after oocyte retrieval; and (4) complications with malignant tumors or other systemic diseases, such as an active stage of systemic lupus erythematosus, that were not suitable for pregnancy.

The research was approved by the Research Ethical Committee of the Second Hospital Affiliated to Wenzhou Medical University (approval number: L-2020-09).

Study groups

In order to determine the effect of peripheral blood estrogen levels on clinical, maternal, and fetal perinatal outcomes of fresh SBT cycles during an early follicular phase prolonged protocol on the day of trigger, we divided the patients who met the inclusion standards into four groups according to their serum E2 concentration amounts. We then used the PERCENTILE function in Microsoft Excel to calculate the corresponding E2 value percentiles. The four groups were: the <25th percentile group (E2 level: 212–1,677 pg/ml, n = 194), the 25th–50th percentile group (E2 level: 1,680–2,380 pg/ml, n = 195), the 51st–75th percentile group (E2 level: 2,407–3,028 pg/ml, n = 188), and the >75th percentile group (E2 level: 3,036–6,526 pg/ml, n = 194).

Ovarian stimulation and embryo culture

We injected a single full dose of 3.75 mg GnRH-a (Triptorelin, Kilferin, Germany) during the D1-D3 menstrual cycle. The ovarian inhibition situation was evaluated 32 and 38 days after pituitary down-regulation. If there were no follicles greater than 8 mm, the E2 level did not exceed 50 pg/ml, and the LH level was less than 5 IU/L, we determined that the pituitary desensitization was successful. Additionally, we used 87.5-375 IU of recombinant human FSH (rhFSH; Gonal-F, Merck Serono, Geneva, Switzerland) for COH according to the patient’s age, BMI, antral follicle count (AFC), and levels of LH, P4, and E2 on the starting day.

We used timely ultrasonic and blood examination to monitor the patients’ follicular growth, ovarian response, peripheral serum E2 levels, and LH and P4 values. If the LH level did not exceed 0.5 IU/L and the follicular growth speed was not ideal, we added 75 IU of recombinant LH (r-LH; Merck Serono, Aubonne, Switzerland). When the average diameter of the two follicles reached 18 mm, we administered a single dose of 4,000–10,000 IU of human chorionic gonadotropin (hCG, Guangdong Lizhu, China). Egg harvesting was performed after 34–36 h.

The embryo culture program has been previously described and the embryo scoring standard referred to the 2011 Istanbul Consensus (Embryology ASiRMaESIGo, 2011; Ying et al., 2019). A blastomere with seven to nine cells of an A or B grade was considered a good quality D3 blastomere. Blastocysts above 4BB of D5 or D6 were considered good quality, and D5 blastocysts that reached at least 3BC could be freshly transferred (Fig. 2).

Figure 2 Prolonged protocol.

One single full dose 3.75 mg GnRH-a was injected during the menstrual cycle of D1–D3. Ovarian inhibition situation was evaluated 32rd to 38th day after pituitary down-regulation. Controlled ovarian hyperstimulation will begin after ovarian inhibition situation evaluation. When the average diameter of two follicles reached 18 mm, a single dose of human chorionic gonadotropin 4,000–10,000 IU was given and egg harvesting was performed after 34–36 h. Fresh SBT will be done after oocyte retrieval. β-hCG test will be done after fresh SBT 12 days later. The first transvaginal ultrasound was taken to evaluate the embryo implantation after the 26th day of fresh SBT, and the second ultrasound was on the 40th day after fresh SBT.

Luteal phase support

Patients were given a 40 mg tablet of Duphaston (Dydrogesterone, Abbott Biologicals, Olst, Netherlands) to be taken orally daily and 200 mg progesterone soft capsules (Utrogestan; Capsugel, Ploermel, France) each day following egg collection. The first transvaginal ultrasound was given to evaluate embryo implantation after the 26th day of fresh SBT, and we reduced the dose of luteal phase support drugs accordingly. The second ultrasound was given on the 40th day after SBT and the dose was reduced according to the patient’s condition. We used a maintenance dose until 2 to 4 weeks had passed after the second ultrasound test (Fig. 2). The various rate calculation methods are shown in Table S1.

Outcome assessment

We confirmed biochemical pregnancy using the serum β-hCG levels, which needed to be at least 10 IU/L on the 12th day following SBT. A clinical pregnancy was determined by the detection of an intrauterine gestational sac via transvaginal ultrasound 26 days after transfer. We considered an ongoing pregnancy after 12 weeks of clinical pregnancy. Early miscarriage occurred before 12 gestational weeks. Moreover, ectopic pregnancy was indicated by ultrasonographic findings of the ectopic gestational sac and/or primitive cardiac tube pulsation. Additionally, we defined monozygotic twins as two fetuses sharing the same gestational sac.

A preterm delivery was regarded as occurring between gestational weeks more than 28 and less than 37. A pregnancy with 42 gestational weeks or more was considered a post-term pregnancy. Structural or functional metabolism abnormalities of the embryo or fetus during intrauterine development were considered birth defects. Low birth weight infants were newborns with birth weights less than 2,500 g. Very low birth weight infants weighed less than 1,500 g at birth, and infants with birth weights of more than 4,000 g were considered macrosomia. The various rate calculation methods are shown in Table S1.

Pregnancy complications include gestational diabetes mellitus (GDM, International Classification of Diseases (ICD) 11 code, JA63), placenta previa (ICD 11 code, JA8B), premature rupture of membranes (ICD 11 code, JA89), gestational hypertension (ICD 11 code, JA23), hyperthyroidism (ICD 11 code, 5A0), oligohydramnios (ICD 11 code, JA88), and prenatal hemorrhage (ICD 11 code, JA41), which has a standard diagnosis (Shavit et al., 2017).

Statistics

Continuous variables were delivered using mean values, standard deviation, and dichotomous variables stated by percentage (n). One way ANOVA or was used for the continuous variables in accordance with the homogeneity of normal distribution variance and Tukey’s multiple comparison correction was used for multiple comparisons among groups. When the data was not normally distributed or the variance was uneven, Kruskal Wallis H test of K-independent sample test in non-parametric test will be used and Mann–Whitney U test or Holm correction were then used for multiple comparisons. P value less than 0.05 indicated that the difference was statistically significant.

Univariate logistic regression analysis was used to test the correlation of related variables including BMI, LH level, and the AFC on the commencing day, etc., and live birth rate. A forest map was drawn to see whether the above confounding factors affect the live birth rate. Receiver operating characteristic (ROC) curves were used to show whether serum E2 levels on the day of trigger could be used to predict the rates of blastocyst implantation, clinical pregnancy, ongoing pregnancy, early abortion, live birth, and early-onset severe OHSS.

The statistical analysis of all data was performed using GraphPad Prism Software (version 8.0.2, San Diego, CA) and SPSS Software (version 22.0; IBM, Armonk, NY, USA).

Results

In this study, we analyzed 771 fresh SBT cycles under an early follicular phase prolonged protocol with β-hCG values no less than 10 IU/L. We compared the patients’ clinical outcomes and perinatal and obstetrical complications according to the serum E2 levels on the day of trigger, baseline information, COH, and laboratory parameters. Baseline information was similar across the four groups, except with regards to BMI; BMI decreased as E2 levels increased on the day of trigger (Table 1).

Table 1 Baseline characteristics of four groups.

	E2 levels on the day of hCG of different groups (pg/ml)	
	<25th percentile Group	25–50th percentile Group	51–75th percentile Group	>75th percentile Group	P value1	
	212–1,677 (n = 194)%	1,680–2,380 (n = 195)%	2,407–3,028 (n = 188)%	3,036–6,526 (n = 194)%		
Maternal age (year)	30.29 ± 4.24	30.24 ± 4.05	30.01 ± 3.77	29.60 ± 3.62	0.296	
Paternal age (year)	33.07 ± 4.77	32.18 ± 4.35	32.51 ± 4.47	32.01 ± 4.67	0.106	
Maternal BMI (kg/m2)	22.44 ± 3.27b, c	22.02 ± 3.25	21.61 ± 2.91	20.96 ± 2.55e, f	0.000	
Infertility types						
primary infertility %(n)	40.21 (78/194)	44.1 (86/195)	41.49 (78/188)	48.45 (94/194)	0.372	
Secondary infertility %(n)	59.79 (116/194)	55.9 (109/195)	58.51 (110/188)	51.55 (100/194)	0.372	
Infertile duration (year)	3.41 ± 2.56	3.20 ± 2.35	3.19 ± 2.25	2.94 ± 2.31	0.283	
Infertile causes					0.346	
female infertility %(n)	46.39 (90/194)	44.10 (86/195)	43.09 (81/188)	43.81 (85/194)		
PCOS (n/n)	(31/90)	(31/86)	(33/81)	(32/85)		
EMs (n/n)	(14/90)	(13/86)	(6/81)	(14/85)		
male infertility %(n)	26.80 (52/194)	26.15 (51/195)	23.93 (45/188)	21.65 (42/194)		
both infertilities %(n)	20.10 (39/194)	25.64 (50/195)	29.79 (56/188)	30.93 (60/194)		
unknown reason %(n)	6.71 (13/194)	4.11 (8/195)	3.19 (6/188)	3.61 (7/194)		
Basal LH (IU/L)	4.74 ± 2.96	4.96 ± 2.84	5.42 ± 3.37	5.45 ± 3.05	0.054	
Basal FSH (IU/L)	7.29 ± 2.02	7.06 ± 1.87	7.00 ± 1.72	6.91 ± 1.68	0.204	
Basic E2 level (pg/ml)	45.90 ± 15.30	48.32 ± 17.42	47.23 ± 17.03	47.90 ± 16.89	0.502	
Notes:

1 P < 0.05 was statistical significance “a” represents P value less than 0.05 between groups 1 and 2, “b” represents P value less than 0.05 between groups 1 and 3, “c” represents P value less than 0.05 between groups 1 and 4, “d” represents P value less than 0.05 between groups 2 and 3, “e” represents P value less than 0.05 between groups 2 and 4, “f” represents P value less than 0.05 between groups 3 and 4.

PCOS, Polycystic ovary syndrome; EMs, Endometriosis; LH, Luteinizing hormone; FSH, Follicle stimulating hormone; E2, Estradiol.

When performing COH under the prolonged protocol, the >75th percentile group had the highest LH level and AFC on the commencing day (0.48 ± 0.21 IU/L and 19.10 ± 6.26, respectively; all P < 0.001). The 25th–50th percentile group had the highest E2 level on the commencing day and the days of stimulation were highest (28.19 ± 10.22 pg/ml, P = 0.003 and 11.40 ± 2.48 days, P < 0.001). The <25th percentile group had the highest P4 level on the commencing day (0.52 ± 0.32 ng/ml P < 0.001). LH levels on the day of trigger were highest in the 51st–75th percentile group (0.81 ± 0.62 IU/L, P < 0.001). The >75th percentile group had the highest levels of total E2, E2 per follicle, P4 and endometrial thickness on the day of trigger (3,992.8 ± 724.4 pg/ml, 259.68 ± 101.14 pg/ml, and 0.85 ± 0.34 ng/ml, 11.42 ± 2.06 mm, respectively; all P < 0.001). However, initial and total Gn doses were highest in the <25th percentile group (201.66 ± 59.03 IU and 2,434 ± 929.88 IU, respectively; all P < 0.001) (Table 2).

Table 2 Controlled ovarian hypersimulation procedure.

	E2 levels on the day of hCG of different groups (pg/ml)		
	<25th percentile Group	25–50th percentile Group	51–75th percentile Group	>75th percentile Group	1P value	
	212–1,677 (n = 194)	1,680–2,380 (n = 195)	2,407–3,028 (n = 188)	3,036–6,526 (n = 194)		
On the commencing day						
LH level (IU/L)	0.38 ± 0.20a, b, c	0.44 ± 0.23d, e	0.42 ± 0.21	0.48 ± 0.21f	0.000	
E2 level (pg/ml)	27.97 ± 11.09a, b	28.19 ± 10.22e	27.88 ± 9.33	27.44 ± 9.70f	0.003	
P4 level (ng/ml)	0.52 ± 0.32c	0.49 ± 0.21e	0.49 ± 0.23	0.47 ± 0.23f	0.000	
AFC (n)	16.02 ± 6.13a, b, c	17.05 ± 6.76d, e	18.42 ± 6.74	19.10 ± 6.26f	0.000	
Ovarian cyst formation rate %(n)	7.22 (14/194)	7.69 (15/195)	5.85 (11/188)	7.73 (15/194)	0.881	
On the day of hCG						
LH level (IU/L)	0.57 ± 0.61a, b, c	0.68 ± 0.59d, e	0.81 ± 0.62	0.81 ± 0.60f	0.000	
Total E2 level (pg/ml)	1,157.46 ± 395.92a, b, c	2,069.63 ± 198.24d, e	2,719.01 ± 173.75	3,992.80 ± 724.40f	0.000	
E2 level per each follicle (pg/ml)	121.20 ± 66.29a, b, c	166.04 ± 53.31d, e	197.15 ± 75.66	259.68 ± 101.14f	0.000	
P4 level (ng/ml)	0.53 ± 0.28a, b, c	0.67 ± 0.32d, e	0.76 ± 0.29	0.85 ± 0.34f	0.000	
Endometrial thickness (mm)	11.19 ± 2.38c	11.28 ± 2.24e	11.10 ± 2.23	11.42 ± 2.06f	0.000	
Initial dose of FSH (IU)	201.66 ± 59.03a, b, c	187.27 ± 56.58d, e	184.13 ± 55.24	177.23 ± 52.32f	0.000	
Days of stimulation (days)	11.29 ± 2.68a, b	11.40 ± 2.48d	11.05 ± 2.12	11.36 ± 2.18f	0.000	
Total dose of FSH (IU)	2,434.97 ± 929.88a, b, c	2,331.68 ± 940.72d, e	2,153.89 ± 796.71	2,123.89 ± 736.30	0.000	
Notes:

1 P < 0.05 was statistical significance “a” represents P value less than 0.05 between groups 1 and 2, “b” represents P value less than 0.05 between groups 1 and 3, “c” represents P value less than 0.05 between groups 1 and 4, “d” represents P value less than 0.05 between groups 2 and 3, “e” represents P value less than 0.05 between groups 2 and 4, “f” represents P value less than 0.05 between groups 3 and 4.

LH, Luteinizing hormone; E2, Estradiol; P4, Progesterone; AFC, Antral Follicle Count.

The laboratory results showed that the >75th percentile group had the highest number of retrieved oocytes, double pronuclear (2PNs), cleavage embryos, D3 embryos, good quality D3 embryos, culture D3 embryos into blastocysts number, blastocysts, and good quality blastocysts per cycle (17.79 ± 4.86, 13.47 ± 4.11, 13.26 ± 4.17, 13.73 ± 4.23, 6.91 ± 3.19, 12.20 ± 4.15, 7.37 ± 2.85, and 3.84 ± 2.61, respectively; all P < 0.001) (Table 3). The rate of good quality D3 embryos was highest in the <25th percentile group (0.53 ± 0.21, P < 0.001). Conversely, the rates of blastocyst formation and good quality blastocyst formation were highest in the 25th–50th percentile group (0.60 ± 0.19, P = 0.017; 0.34 ± 0.19, P = 0.037) (Table 3). Our comparisons of the grades of blastocyst cavity expansion, inner cell mass, and trophoblastic layer showed no significant differences. Blastocyst cavity expansion was almost the 4th degree of all the groups. The rate of Grade A or B inner cell masses was similar across the four groups, while the trophoblastic layer was mainly composed of Grade B cells with no statistical difference across the groups (Table 3).

Table 3 Laboratory outcomes of the four groups.

	E2 levels on the day of hCG of different groups (pg/ml)		
	<25th percentile Group	25–50th percentile Group	51–75th percentile Group	>75th percentile Group	P value1	
	212–1,677 (n = 194)	1,680–2,380 (n = 195)	2,407–3,028 (n = 188)	3,036–6,526 (n = 194)		
Fertilization methods					0.421	
IVF % (n)	80.93 (157/194)	80.00 (156/195)	76.60 (147/188)	78.87 (153/194)		
ICSI % (n)	14.95 (29/194)	14.87 (29/195)	18.62 (35/188)	19.59 (38/194)		
IVF/ICSI % (n)	4.12 (8/194)	5.13 (10/195)	3.19 (6/188)	1.54 (3/194)		
No. of oocytes retrieved (n)	12.83 ± 4.69a, b, c	15.41 ± 5.17d, e	16.83 ± 4.82	17.94 ± 4.86f	0.000	
No. of 2PNs per cycle (n)	9.67 ± 4.05a, b, c	11.82 ± 4.05d, e	12.87 ± 4.68	13.47 ± 4.11f	0.000	
No. of cleavage per cycle (n)	9.50 ± 3.98a, b, c	11.52 ± 4.01d, e	12.60 ± 4.60	13.26 ± 4.17f	0.000	
No. of D3 embryos per cycle (n)	9.87 ± 4.09a, b, c	12.05 ± 4.21d, e	13.14 ± 4.76	13.73 ± 4.23f	0.000	
No. of D3 good quality embryos per cycle (n)	5.11 ± 2.91a, b, c	6.28 ± 3.05e	6.33 ± 3.29	6.91 ± 3.19f	0.000	
Rate of good quality D3 embyros	0.53 ± 0.21b, c	0.52 ± 0.20d, e	0.48 ± 0.19	0.51 ± 0.20f	0.000	
Culture D3 embyros into blastocysts number (n)	8.12 ± 3.72a, b, c	10.55 ± 3.97d, e	11.49 ± 4.30	12.20 ± 4.15	0.000	
No. of D5 and D6 embryos per cycle (n)	5.49 ± 2.89a, b,c	6.93 ± 3.11e	6.96 ± 3.38	7.37 ± 2.85f	0.000	
No. of good quality D5 and D6 embryos per cycle (n)	2.91 ± 2.18a, b, c	3.76 ± 2.07d	3.78 ± 2.71	3.84 ± 2.61	0.000	
Rate of blastocyst formation	0.59 ± 0.22	0.60 ± 0.19d	0.54 ± 0.20	0.57 ± 0.19	0.017	
Rate of good quality blastocyst formation	0.32 ± 0.22	0.34 ± 0.19d, e	0.29 ± 0.20	0.29 ± 0.18	0.037	
Blastocyst condition						
Blastocyst cavity expansion grade					0.791	
Grade 3 blastocyst %(n)	5.67 (11/194)	4.62 (9/195)	5.32 (10/188)	5.67 (11/194)		
Grade 4 blastocyst %(n)	89.18 (173/194)	88.21 (172/195)	90.43 (170/188)	90.72 (176/194)		
Grade ≥5 blastocyst %(n)	5.15 (10/194)	7.18 (14/195)	4.26 (8/188)	3.61 (7/194)		
Inner cell mass grade					0.666	
Grade A %(n)	46.39 (90/194)	50.26 (98/195)	44.68 (84/188)	53.09 (103/194)		
Grade B %(n)	52.06 (101/194)	48.72 (95/195)	53.19 (100/188)	45.88 (89/194)		
Grade C %(n)	1.55 (3/194)	1.03 (2/195)	2.13 (4/188)	1.03 (2/194)		
Trophoblastic layer grade					0.238	
Grade A %(n)	15.46 (30/194)	11.79 (23/195)	17.02 (32/188)	12.87 (25/194)		
Grade B %(n)	76.29 (148/194)	85.13 (166/195)	77.13 (145/188)	80.93 (157/194)		
Grade C %(n)	8.25(16/194)	3.08 (6/195)	5.85 (11/188)	6.19 (12/194)		
Notes:

1 P < 0.05 was statistical significance “a” represents P value less than 0.05 between groups 1 and 2, “b” represents P value less than 0.05 between groups 1 and 3, “c” represents P value less than 0.05 between groups 1 and 4, “d” represents P value less than 0.05 between groups 2 and 3, “e” represents P value less than 0.05 between groups 2 and 4, “f” represents P value less than 0.05 between groups 3 and 4.

Good quality D3 embyros means 7-9, A-B; good quality blastocyst means at least 4BB.

Although the >75th percentile group had the lowest rates of clinical gravidity, embryo transplantation, persistent pregnancy, and living birth, there were no significant differences (85.57% (166/194), 86.60% (168/194), 71.13% (138/194), and 71.13% (138/194), respectively; all P > 0.05). Additionally, rates of severe OHSS, early miscarriage, and ectopic pregnancy showed no significant differences (Table 4).

Table 4 Clinical outcomes of four groups.

	E2 levels on the day of hCG of different groups (pg/ml)		
	<25th percentile Group	25–50th percentile Group	51–75th percentile Group	>75th percentile Group	P value1	
	212–1,677 (n = 194)	1,680–2,380 (n = 195)	2,407–3,028 (n = 188)	3,036–6,526 (n = 194)		
Clinical pregnancy rate %(n)	91.24 (177/194)	88.72 (173/195)	93.06 (175/188)	85.57 (166/194)	0.0869	
Early severe OHSS rate %(n)	0 (0/194)	1.03 (2/195)	1.60 (3/188)	1.55 (3/194)	0.3774	
Embryo implantation rate %(n)	92.78 (180/194)	88.72 (173/195)	93.62 (176/188)	86.60 (168/194)	0.0607	
Biochemical pregnancy rate %(n)	8.76 (17/194)	11.28 (22/195)	6.91 (13/188)	14.43 (28/194)	0.0869	
Ongoing pregnancy rate %(n)	76.80(149/194)	72.31 (141/195)	81.38 (153/188)	71.13 (138/194)	0.0809	
Early miscarriage rate %(n)	9.60 (17/177)	12.14 (21/173)	10.29 (18/175)	14.46 (24/166)	0.5017	
Ectopic pregnancy rate %(n)	0.56 (1/177)	0.58(1/173)	0 (0/175)	0.60(1/166)	0.7953	
Live birth rate per transfer %(n)	76.29 (148/194)	72.31 (141/195)	81.38 (153/188)	71.13 (138/194)	0.0867	
Notes:

1 P-value compared to <25th percentile Group, P < 0.05 was statistical significance.

OHSS, Ovarian hyperstimulation syndrome.

In order to observe the influence of variables with statistical differences among groups on the live birth percentage, we performed a logistic regression analysis. We found that there was no correlation between BMI, LH level, E2 level, P4 level and AFC on the first day of treatment; LH level, total E2 level, E2 level per follicle, P4 level and endometrial thickness on the day of trigger; or the initial and total FSH doses, days of stimulation and number of obtained oocytes, 2PNs, cleavage stage embryos, D3 embryos, good quality D3 embryos, D3 embryos cultured into blastocysts, blastocysts, and good quality blastocysts, rate of good quality D3 embryos, rate of blastocyst formation, rate of good quality blastocyst formation (all P > 0.05, Fig. 3).

Figure 3 Logistic regression of confounding factors.

In order to observe the influence of variables with statistical differences among groups on the live birth percentage, we performed a logistic regression analysis. We found that there was no correlation between BMI, LH level, E2 level, P4 level and AFC on the first day of treatment; LH level, total E2 level, E2 level per follicle, P4 level and endometrial thickness on the day of trigger; or the initial and total FSH doses, days of stimulation and number of obtained oocytes, 2PNs, cleavage stage embryos, D3 embryos, good quality D3 embryos, D3 embryos cultured into blastocysts, blastocysts, and good quality blastocysts, rate of good quality D3 embryos, rate of blastocyst formation, rate of good quality blastocyst formation (all P > 0.05)

We used ROC curves to further analyze the correlation between the E2 levels on the day of hCG administration and the prediction of blastocyst plantation, clinical gravidity, sustained gestation, early abortion, living birth, and early-onset severe OHSS. We found that E2 levels on the day of trigger could not be used to predict the rates of fresh single SBT, clinical pregnancy, sustained pregnancy, early abortion, or live birth, and that the AUC was from 51.33% to 55.82% with P values greater than 0.05. We noted that the E2 level and early onset OHSS ROC curves showed a moderate intensity correlation with an AUC of 72.39%, P value of 0.0292, and a Youden index (YI) of 44.59%; the cut off value for E2 was 2,893 pg/ml with the 75% sensitivity and 70% specificity (Fig. S1).

The >75th percentile group had the highest male-female neonate rate (1.68 (89/53)), but we did not find any significant differences when compared to the other three groups. The 51st–75th percentile group had the highest percentages of low birth weight infants, premature deliveries, hospitalizations in the neonatal intensive care unit (NICU), twin pregnancies, and monochorionic diamniotic pregnancies (11.73% (19/162), P = 0.0408; 11.43% (20/175), P = 0.0269; 10.49% (17/162), P = 0.0029; 8.57% (15/175), P = 0.0047; and 8.57% (15/175); P = 0.001, respectively) (Table 5). Additionally, we found no differences in birth defects across the four groups. Obstetrical complications including GDM, PROM, gestational hypertension, placenta previa, and PPROM also did not show significant differences (Table 5).

Table 5 Comparison of obstetric and perinatal outcomes among different E2 levels.

	E2 levels on the day of hCG of different groups (pg/ml)		
	<25th percentile Group	25–50th percentile Group	51–75th percentile Group	>75th percentile Group	P value1	
	212–1,677 (n = 194)	1,680–2,380 (n = 195)	2,407–3,028 (n = 188)	3,036–6,526 (n = 194)		
Perinatal outcomes						
Gestational weeks (w)	35.21 ± 9.83	34.09 ± 10.8	35.51 ± 9.31	34.77 ± 10.73	0.5897	
Cesarean section rate %(n)	38.98 (69/177)	40.46 (70/173)	48.57 (85/175)	40.96(68/166)	0.2637	
Male/Female neonates ratio (n)	1.04 (79/76)	1.18 (79/67)	1.03 (82/80)	1.68 (89/53)	0.1330	
Birth wright (g)	3,298 ± 589	3,243 ± 606.2	3,141 ± 633.1	3,254 ± 457.2	0.1001	
Low birth wight infant rate %(n)	5.16 (8/155	6.16 (9/146)	11.73 (19/162)	4.23 (6/142) f	0.0408	
Very low birth wight infant rate %(n)	1.94 (3/155)	2.74 (4/146)	2.47 (4/162)	0 (0/142)	0.2932	
Macrosomia rate %(n)	7.74 (12/155)	6.16 (9/146)	3.70 (6/162)	2.82 (4/142)	0.1921	
Premature delivery rate %(n)	3.95 (7/177)b	5.20 (9/173)d	11.43 (20/175)	6.02 (10/166)	0.0269	
Postmature delivery rate %(n)	0 (0/177)	0.58 (1/173)	0 (0/175)	0.60 (1/166)	0.5551	
Admission to NICU rate %(n)	1.93 (3/155)b	5.45 (8/146)	10.49 (17/162)	2.82 (4/142)f	0.0029	
Birth defect rate %(n)	3.87 (6/155)	4.79 (7/146)	2.47 (4/162)	2.11 (3/142)	0.5410	
Twin pregnancy rate %(n)	2.26 (4/177)b	3.47 (6/173)d	8.57 (15/175)	1.81 (3/166)f	0.0047	
Monochorionic diamniotic rate %(n)	1.13 (2/177)b	2.89 (5/173)d	8.57 (15/175)	1.81 (3/166)f	0.0010	
Dichorionic diamnionic rate %(n)	1.13 (2/177)	0.58 (1/173)	0 (0/175)	0 (0/166)	0.3141	
Obstetrics complications						
GDM rate %(n)	8.39 (13/155)	6.85 (10/146)	6.17 (10/162)	2.82 (4/142)	0.2385	
Gestational Hypertension rate %(n)	2.58 (4/155)	3.42 (5/146)	3.70 (6/162)	2.11 (3/142)	0.8376	
Placenta previa rate %(n)	2.58 (4/155)	1.37 (2/146)	2.47 (4/162)	2.82 (4/142)	0.8493	
PROM rate %(n)	1.29 (2/155)	0.68 (1/146)	0 (0/162)	2.11 (3/142)	0.2933	
PPROM rate %(n)	1.29 (2/155)	0.68 (1/146)	2.47 (4/162)	0 (0/142)	0.2205	
*Others rate %(n)	1.29 (2/155)	2.05 (3/146)	2.47 (4/162)	1.41 (2/142)	0.8481	
Notes:

* Others included oligohydramnios, velamentous placenta, hyperthyroidism, battledore placenta, antepartum haemorrhage.

1 P < 0.05 was statistical significance.

NICU, Neonatal intensive care unit; GDM, Gestational diabetes mellitus; PROM, Premature rupture of membranes; PPROM, Preterm premature rupture of membranes.

“a” represents P value less than 0.05 between groups 1 and 2, “b” represents P value less than 0.05 between groups 1 and 3, “c” represents P value less than 0.05 between groups 1 and 4, “d” represents P value less than 0.05 between groups 2 and 3, “e” represents P value less than 0.05 between groups 2 and 4, “f” represents P value less than 0.05 between groups 3 and 4.

Discussion

Few studies have examined the influence of fresh SBT E2 levels on clinical and maternal-fetal perinatal outcomes under a full long acting GnRH-a regimen on the day of trigger. Our results indicated that E2 levels on the day of trigger are not effective when used as a predictor for the clinical and perinatal outcomes of mothers and fetuses. However, we found that E2 levels could still be used to predict early-onset severe OHSS in IVF/ICSI patients with fresh SBT cycles.

There is growing interest in the safety of IVF/ICSI for mothers and newborns, and a great number of doctors and patients in reproductive centers around the world are using SBT because fresh blastocyst stage transfer is a better choice compared to fresh cleavage stage embryo transfer (Glujovsky et al., 2016). No consensus has been reached on the greater success of fresh SBT or vitrified-warmed SBT. Shavit et al. found that vitrified-warmed SBT was associated with poor clinical outcomes and a higher incidence of preeclampsia compared to fresh SBT (Shavit et al., 2017). However, Wei et al. found that frozen SBT resulted in more successful live births in ovulatory women than fresh SBT, but that frozen SBT resulted in an increased risk of pre-eclampsia (Wei et al., 2019). A different study found that there was an equal number of clinical pregnancies and live births using frozen-thawed and fresh blastocyst transfer (Feng et al., 2012). Nonetheless, the majority of reproductive clinicians select the most suitable transplant plan after considering the patient’s safety and the optimal clinical benefits.

Prolonged pituitary down-regulation before starting Gn resulted in higher live-birth rates during fresh embryo transfer due to the greater endometrial receptivity during embryo implantation (Ren et al., 2014; Tian et al., 2019). In our previous study, we found one GnRH-a depot across different fresh embryo transfer menstrual cycles, which was consistent with the findings of Ying et al. (2019). However, few studies have further explored how E2 levels influence fresh SBT’s clinical and maternal-fetal perinatal outcomes under this protocol.

During the fresh embryo transfer scheme, the hyper-physiological levels of E2 elevated the P4 levels, which reduced endometrial receptivity (Wu et al., 2012). Li et al. confirmed that in IVF/ICSI patients, the supra-physiological E2 concentration in COH affected endometrial transcriptome profiles, resulting in a shift of the embryo transfer window via endometrial mRNA and lncRNA sequencing (Li et al., 2020). Other researchers found that there was not sufficient evidence to confirm or deny the effect of E2 levels on clinical outcomes during the IVF/ICSI cycle on the day of hCG administration (Kosmas, Kolibianakis & Devroey, 2004). Although the basic BMI information was biased across the four groups, this information did not affect our logistic regression analysis that calculated the patients’ live birth rates.

In this study, the >75th percentile group had the highest LH level and AFC on the commencing day, total E2, E2 per follicle, P4 levels and endometrial thickness on the day of trigger. Table 1 shows that, although we found no statistical differences in the basic endocrinology across the four groups, the group with the highest E2 level (the >75th percentile group) had the lowest average age, basal FSH level, and FSH/LH ratio, indicating that this group may have better ovarian reserve function (Lensen et al., 2018). Therefore, patients with the highest E2 levels had more eggs with the minimum initial and total Gn amount. Additionally, the >75th percentile group had the highest number of retrieved oocytes, 2PNs, cleavage embryos, D3 embryos, good quality D3 embryos, D3 embryos cultured into blastocysts, blastocysts, good quality blastocysts per cycle. While, the 25th–50th percentile group had the highest rates of blastocyst formation and good quality blastocyst formation. Though we saw no statistical differences in the clinical outcomes and P values ranged from 0.06 to 0.08, the cluster with the highest E2 levels had the lowest rates of blastocyst embeddedness, clinical and ongoing pregnancy, and live birth. It is likely that statistical differences will be observed if the sample size increases.

In this study, we mapped the relationship between E2 levels and six different clinical outcomes. Although the ROC results could not predict the clinical outcomes, the E2 level on the day of trigger was still a relatively reliable indicator of early-onset severe OHSS, an E2 cut-off value of 2,893 pg/ml with the 75% sensitivity and 70% specificity (Magnusson et al., 2018). The mean E2 level for all patients in the study was 2,483 pg/ml. The span between this value and the E2 cut-off value for early-onset severe OHSS was 410 pg/ml. Even though the incidence of early-onset severe OHSS was low (ranging from 0–1.6%) in regards to this prolonged regimen, clinicians should strictly measure Gn dosages during the final stage of COH for patients with good ovarian function, and should be cautious when increasing the Gn dosage in order to better control the risk of early-onset severe OHSS.

Previous studies have confirmed that retrieving 11–15 oocytes produces satisfactory live birth rates and cumulative live births with the lowest rate of OHSS, while retrieving more than 15 oocytes can critically increase the risk of OHSS without increasing the live birth rate (Malchau et al., 2019; Steward et al., 2014). In this study, the average number of eggs retrieved in the <25th, 25th–50th, and 51st–75th groups was between 11.04 and 15.44, and the clinical outcomes were encouraging. This result was in accordance with the previous studies, suggesting that a prolonged protocol combined with fresh SBT would be beneficial for most suitable patients.

The results of this study suggested that the twin conception rate from SBT was much less than that of two blastomeres transfer, which was consistent with the results of a previous study (De Croo et al., 2019). Although SBT reduces the rate of twin pregnancy, it increased the incidence of monozygotic twin pregnancies (Busnelli et al., 2019; Liu et al., 2018; Mateizel et al., 2016), which we observed in our study, especially in the 51st–75th percentile group (8.57% (15/175), P = 0.0047). The incidence of monozygotic twins is 0–13.2% during blastocyst transplantation (Hviid et al., 2018), and another study showed that the incidence of multiple pregnancies with zygotic division after single embryo transfer is 1.36% (Ikemoto et al., 2018). Nevertheless, there is no consensus on why the monozygotic twinning rate increases during blastocyst transfer, but the possible explanations are the impact of maternal age, extended culture in fresh and frozen cycles, fresh transfer, in vitro maturation, assisted hatching, and the treatment year (Busnelli et al., 2019; Ikemoto et al., 2018; Knopman et al., 2014; Liu et al., 2018; Mateizel et al., 2016; Ota et al., 2020). Other studies have shown that the monozygotic twin rate is unrelated to maternal age, the use of diverse zona pellucida operation techniques, the type of culture medium, ovarian stimulation, or fertilization methods (Ikemoto et al., 2018; Mateizel et al., 2016).

The mechanism of monozygotic twinning is not fully understood. The accepted theory is that the embryo splits after fertilization to produce monozygotic twins (Sundaram, Ribeiro & Noel, 2018). Corner’s model suggested that herniation and subsequent splitting of the inner cell mass during the blastocyst stage could produce monozygotic twins (Knopman et al., 2010; Nakasuji et al., 2014). Some scholars believe that prolonging embryo culture time could reduce intercellular adhesion and lead to the division of the inner cell mass (Behr et al., 2000). Embryo splitting occurs after the blastocyst stage (Kyono, 2013). It has also been suggested that the occurrence of monozygotic twins could be reduced by using the time-lapse technique to exclude blastocysts that contain loose inner cell mass (Otsuki et al., 2016). We hypothesize that the incidence of monozygotic twins was so high (between 2,407 and 3,028 pg/ml) on the day of trigger in our experiment because the concentration of E2 may have destroyed the adhesion of the inner cell mass. However, this needs to be further explored.

Conclusion

By analyzing the effect of E2 levels in fresh SBT under a prolonged protocol on clinical, maternal, and fetal perinatal outcomes on the day of trigger, we found that E2 concentration may not be effective at predicting patients’ clinical and perinatal outcomes. However, E2 levels can still be used to predict the incidence of early-onset severe OHSS, and the cut-off value was 2,893 pg/ml with the 75% sensitivity and 70% specificity under a prolonged protocol. When estrogen levels were between 2,407 and 3,028 pg/ml, we found the highest proportion of single chorionic twins transferred with fresh SBT, leading to an increase in premature births, NICU hospitalizations, and low birth weight. Overall, using a prolonged protocol with fresh SBT offers patients efficiency and security. It may be worthwhile to design a prospective study that can further investigate the effect of trigger day E2 levels on the clinical and perinatal effects of fresh SBT under an early follicular phase prolonged protocol. It would also be valuable to explore the mechanism of E2 concentration on the day of trigger in the incidence of monozygotic twins.

Supplemental Information

Supplemental Information 1 Raw data.

Click here for additional data file.

Supplemental Information 2 ROC curves analysis of the correlation between E2 level and clinical outcomes.

ROC was used to further analyze the correlation between E2 level on the trigger day and forecasts of fresh single SBT, clinical pregnancy, sustained pregnancy, early abortion, live birth and early-onset severe OHSS. We found that the level of E2 on the trigger day was unpredictable of rates of fresh single SBT, clinical pregnancy sustained pregnancy, early abortion and live birth, that the AUC were from 51.33% to 55.82% with P value greater than 0.05 (Shown in A-E). What is noteworthy is that ROC curve of E2 level and early set-on OHSS had moderate intensity correlation, of which AUC was 72.39%, P value was 0.0292, youden index (YI) was 44.59% and the cut off value of E2 was 2,893 pg/ml with the 75% sensitivity and 70% specificity (Shown in F).

Click here for additional data file.

Supplemental Information 3 Definitions of different rates.

Click here for additional data file.

Additional Information and Declarations

Competing Interests

Author Contributions

Human Ethics

Data Availability

The authors declare that they have no competing interests.

Yingfen Ying conceived and designed the experiments, performed the experiments, analyzed the data, prepared figures and/or tables, authored or reviewed drafts of the paper, and approved the final draft.

Xiaosheng Lu performed the experiments, prepared figures and/or tables, and approved the final draft.

Huina Zhang performed the experiments, prepared figures and/or tables, and approved the final draft.

Samuel Kofi Arhin analyzed the data, prepared figures and/or tables, and approved the final draft.

Xiaohong Hou analyzed the data, authored or reviewed drafts of the paper, and approved the final draft.

Zefan Wang analyzed the data, authored or reviewed drafts of the paper, and approved the final draft.

Han Wu analyzed the data, authored or reviewed drafts of the paper, and approved the final draft.

Jieqiang Lu conceived and designed the experiments, authored or reviewed drafts of the paper, and approved the final draft.

Yunbing Tang conceived and designed the experiments, authored or reviewed drafts of the paper, and approved the final draft.

The following information was supplied relating to ethical approvals (i.e., approving body and any reference numbers):

Research Ethical Committee of the 2nd Hospital Affiliated to Wenzhou Medical University approved this research (approval number: L-2020-09).

The following information was supplied regarding data availability:

The raw measurements are available as Supplemental File 1.

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
