# Peer review of "Clinical and perinatal outcomes of fresh single-blastocyst-transfer cycles under an early follicular phase prolonged protocol according to day of trigger estradiol levels"

_PeerJ, doi:10.7717/peerj.11785_

## Round 0.1 · original submission · Minor Revisions

· Academic Editor

Minor Revisions

The elaborate and labor-intensive work provides the valuable clinical observation in regard with fresh single-blastocyst-transfer (SBT) and estradiol level on the day of trigger. However, statistical anaylsis needs to be improved as the reviewer mentioned. After minor revisions according to the reviewers' comments, this manuscript would be accepted.

Reviewer 1 ·

Basic reporting

1. Table1 –Table5, the footnotes need to be revised. It was unclear what tests were performed to get these p-values and why some of the groups has more than one p-value. These information should be added to the footnote and the main text. In table3, the numbers for ‘Rate of blastocyst formation’ do not seem right. For continuous variables, it would be better to add the statistics listed in the table, eg.maternal age, mean ± SD.
2. Figure3 does not have titles or labels.

Experimental design

1. In the statistics section, the authors listed all the tests; however, it was not clear enough when the tests were performed. To perform a one-way ANOVA test, there is no need to specify the reference group.
2. In line 194-195, when stating multiple comparisons corrections, it is not appropriate to say it is the ‘comparison between continuous and dichotomous variables’, instead, using something similar like ‘Tukey’s multiple comparison correction was used for pairwise comparison among groups’. In addition, fisher’s exact test or Chi-sq test is not the appropriate test for multiple comparisons for categorical variables; the authors should consider using other corrections like Holm correction. Moreover, can the authors explain the reason why the criteria of statistical significance was set to p<0.008?
3. In line 197, the authors stated that ‘multiple regression analysis was used to test the correlation of related variables’, however, it was unclear what type of regression was performed here, and it is confusing about the purpose of performing the regression. If it is to test for the correlation between variables, Spearman’s correlation or Pearson’s correlation coefficient should do the work. If it is to determine the relationship between live birth and its predictors, univariate logistic regression is suitable, but the wording should be revised.
4. The authors utilized ROC curve to see if E2 levels could be used to predict outcomes, however, this is not appropriate since a Receiver operating characteristics only tells you how the sensitivity and specificity will trade off, and should be thought of as supplemental to the main analysis, rather than the primary analysis itself. The authors might want to fit logistic regression models for these outcomes and calculate the cutoffs based on the predicted probability. The corresponding results should also be revised based on the new results.
5. It would be interesting if the authors can list the sensitivity and specificity achieved on that optimal cut-off point.
6. In line 198, it seems like the authors performed logistic regressions to calculate the ORs and their 95% CIs, however, it would be better if the authors not use the ‘rate’ or ’percentage’ word as the outcome, otherwise it will be confusing if this is a continuous rate outcome or binary outcome. This language should be revised throughout the paper.

Validity of the findings

The results after revising the statistical method should be updated accordingly.

Additional comments

This is an engaging article with robust methodology that purposefully questions our knowledge of
the subject, however, further revision is required for this manuscript.

Reviewer 2 ·

Basic reporting

no comment

Experimental design

In a multivariate system, often one variable may not be able to predict the system accurately. In the future studies, it would be beneficial to build a predictive model using:

1. more sophisticated modeling approaches capable of capturing non-linear relationships
2. multiple variables

Validity of the findings

no comment

Additional comments

This is a very well written manuscript describing the prediction accuracy of E2 level on various pregnancy characteristics of patients in single-blastocyst-transfer cycles. The results are presented clearly and the scientific English language has added to the strength of the manuscript. This study provides invaluable experimental data for researchers to build more sophisticated models such as neural networks in order to predict different pregnancy factors. For this reason, I highly recommend that the experimental data to be publicly available if possible.

---

## Round 0.2 · accepted · Accept

· Academic Editor

Accept

Thank you for sending PeerJ the revised manuscript. Authors provide answers accordingly upon reviewers' comments. Therefore, the current form is suitable to be accepted as it stands.

Reviewer 1 ·

Basic reporting

no comment

Experimental design

no comment

Validity of the findings

no comment

Additional comments

Thanks the authors for addressing my questions. The current version looks great to me.